# Heart Rate Variability Measurement Can Be a Point-of-Care Sensing Tool for Screening Postpartum Depression: Differentiation from Adjustment Disorder

**DOI:** 10.3390/s24051459

**Published:** 2024-02-23

**Authors:** Toshikazu Shinba, Hironori Suzuki, Michiko Urita, Shuntaro Shinba, Yujiro Shinba, Miho Umeda, Junko Hirakuni, Takemi Matsui, Ryo Onoda

**Affiliations:** 1Department of Psychiatry, Shizuoka Saiseikai General Hospital, Shizuoka 422-8527, Japan; 2Research Division, Saiseikai Research Institute of Health Care and Welfare, Tokyo 108-0073, Japan; 3Autonomic Nervous System Consulting, Shizuoka 420-0839, Japan; 4Department of Obstetrics and Gynecology, Shizuoka Saiseikai General Hospital, Shizuoka 422-8527, Japan; 5Department of Obstetrics and Gynecology, Nagoya University Graduate School of Medicine, Nagoya 466-8550, Japan; 6Ward North 3, Shizuoka Saiseikai General Hospital, Shizuoka 422-8527, Japan; 7School of System Design, Tokyo Metropolitan University, Tokyo 191-0065, Japan

**Keywords:** heart rate variability, three-behavioral-state, postpartum depressive disorder, adjustment disorder, screening, point-of-care sensing tool

## Abstract

Postpartum depression (PPD) is a serious mental health issue among women after childbirth, and screening systems that incorporate questionnaires have been utilized to screen for PPD. These questionnaires are sensitive but less specific, and the additional use of objective measures could be helpful. The present study aimed to verify the usefulness of a measure of autonomic function, heart rate variability (HRV), which has been reported to be dysregulated in people with depression. Among 935 women who had experienced childbirth and completed the Edinburgh Postnatal Depression Scale (EPDS), HRV was measured in EPDS-positive women (*n* = 45) 1 to 4 weeks after childbirth using a wearable device. The measurement was based on a three-behavioral-state paradigm with a 5 min duration, consisting of rest (Rest), task load (Task), and rest-after-task (After) states, and the low-frequency power (LF), the high-frequency power (HF), and their ratio (LF/HF) were calculated. Among the women included in this study, 12 were diagnosed with PPD and 33 were diagnosed with adjustment disorder (AJD). Women with PPD showed a lack of adequate HRV regulation in response to the task load, accompanying a high LF/HF score in the Rest state. On the other hand, women with AJD exhibited high HF and reduced LF/HF during the After state. A linear discriminant analysis using HRV indices and heart rate (HR) revealed that both the differentiation of PPD and AJD patients from the controls and that of PPD patients from AJD patients were possible. The sensitivity and specificity for PPD vs. AJD were 75.0% and 90.9%, respectively. Using this paradigm, an HRV measurement revealed the characteristic autonomic profiles of PPD and AJD, suggesting that it may serve as a point-of-care sensing tool in PPD screening systems.

## 1. Introduction

Postpartum depression (PPD) profoundly affects the lives of mothers and their families after childbirth [1,2]. Women show symptoms of major depressive disorder, including depressed feelings, anhedonia, and irritability, with the onset of PPD after childbirth [3]. The risk of suicide is also high in PPD patients, leading to social problems [4]. However, the literature suggests that diagnostic processes are inconsistent, depending on the medical and social environment. The reported prevalence rates of PPD are different across the available studies, varying from 4.4% to 73.7% after delivery [5], which suggests that PPD in the present diagnostic system may not be a homogeneous disorder. Cross-cultural differences in prevalence rates across many countries have also been reported [6,7]. An accurate diagnosis of PPD among women after childbirth, especially in areas where psychiatrists are not present, is important in order to start adequate treatment early.

Screening by means of questionnaires has been found to be effective in detecting the presence of PPD [8]. Women who are screened and found to be positive for PPD can reach out for supporting resources. Among the various questionnaires available, the Edinburgh Postnatal Depression Scale (EPDS) has been widely utilized [9], and its cut-off point has been reported to be sensitive for screening PPD [10]. However, regarding the specificity of the available questionnaires, including the EPDS, previous reports have indicated their limitation in differentiating PPD from other psychiatric disturbances, including adjustment disorder (AJD) [11].

AJD is a disorder with marked distress following stress exposure and may significantly impair social functioning [3]. Childbirth can be a stressor leading to AJD. Its treatment is different from that for PPD, including psychological and environmental assessment. The differentiation of PPD and AJD is important to adequately start the care of women with mental health issues. In a study using a cut-off point of 9, the EDPS-positive rate was 14.8%, while the incidence of postpartum depression was 5.4% [10]. Ferrari et al. [12] reported that among the women screened in their study, 14.6% of those screened were positive, but 23.6% of the women suffered from AJD and 5.5% suffered from PPD. Therefore, PPD and AJD could not be distinguished based on EPDS scores only [11]. It is necessary to increase the specificity of current screening systems for PPD by enabling the differentiation of PPD from other psychiatric disorders.

PPD is a biopsychosocial disturbance involving not only psychosocial factors, such as interpersonal relationship and perinatal stress [13,14,15], but also biological factors, including hormonal, physical, microbiological, metabolic, genetic, and physiological conditions [16,17,18,19,20,21]. To increase the specificity of screening systems for a proper diagnosis, it may be necessary to add biological measures. Among the biological candidates, hormonal changes, including estrogen, oxytocin, and corticotropin-releasing hormone levels, have been extensively studied, and both positive and negative results have been reported in relation to the risk of PPD [22,23,24,25,26,27]. Genetic factors have also been investigated as markers for PPD [28]. As for the physiological index, heart rate variability (HRV) has been investigated [29], and Solorzano et al. [30] reported that low time-domain HRV in the prepartum period was related to high depressive symptoms in the postpartum period.

Among the biological candidates, it may be revealing to add HRV measurement to current screening systems, because the results of HRV analysis can be obtained promptly after its measurement without delay, together with the results of the questionnaires. Wearable devices can be utilized, enabling HRV measurement not only in clinic offices but also in counseling rooms and in women’s own homes. HRV measurement could be suitable as a point-of-care tool for diagnostic aid because some of the HRV data are obtained automatically, and both medical staff and mothers themselves can utilize the information to consider the course of treatment.

HRV also has another advantage as a parameter to use for the diagnosis of PPD: it has been widely studied to evaluate autonomic changes in major depressive disorder [31]. Variation in inter-heartbeat intervals is called HRV, and reflects the activity of the autonomic nervous system [32]. Both time-domain and frequency-domain HRV indices are used to evaluate the activity of the autonomic nervous system [33]. HRV was originally used to assess cardiac function in coronary heart diseases, and at present, is extensively employed to examine autonomic function in psychiatric disorders in addition to various somatic disorders. Our previous studies have revealed that the use of a three-behavioral-state paradigm during frequency-domain HRV measurement is not only effective for detecting major depressive disorder but also for differentiating it from other disorders, including anxiety disorder and chronic fatigue syndrome [34,35]. The paradigm can be conducted at an outpatient office in 5 min and causes little distress to the patients. The present study examined the usefulness of HRV measurement during this three-behavioral-state paradigm using a wearable electrocardiography device as a point-of-care sensing tool for screening PPD in addition to the use of EPDS.

## 2. Materials and Methods

### 2.1. Participants

A total of 935 women who had a experienced childbirth during the period between September 2017 and February 2019 at Shizuoka Saiseikai General Hospital participated in the present study and completed the EPDS, which consists of 10 questions, 1 to 4 weeks after childbirth [9]. A score of 9 or more was regarded as EPDS-positive according to the data of previous studies [10]. Among those who presented an EPDS-positive score (*n* = 58, 6.1% of the total women), 45 women without a history of psychiatric, neurological, cardiovascular, arrhythmic, and metabolic illnesses were enrolled in the present HRV measurement protocol (4.8%, age: 32.6 ± 5.2 years, mean ± s.d.). The EPDS-positive women excluded from enrollment in the present study included those who had been treated for depression or anxiety disorder before childbirth. HRV measurement was conducted in an outpatient office using a wearable device, as presented below.

On the day of HRV measurement, a psychiatric diagnosis was made based on the criteria of the Diagnostic and Statistical Manual of Mental Disorders, 5th edition (DSM-5) [3] by a psychiatrist certified by the Japanese Society for Psychiatry and Neurology. Twelve women were diagnosed with major depressive disorder with postpartum onset (PPD, 26.7% of the 45 EDPS-positive women, 1.3% of the total women after childbirth, age: 34.4 ± 4.1 years, parity; 1.3 ± 0.8, gestational age of the child; 258.3 ± 27.9 days, body weight of the child at birth; 2607.9 ± 641.1 g). The remaining women were diagnosed with adjustment disorder (AJD, *n* = 33, 73.3% of the 45 EDPS-positive women, 3.5% of the total women after childbirth, age: 32.6 ± 5.5 years, parity; 1.3 ± 0.4, gestational age of the child; 262.4 ± 17.0 days; body weight of the child at birth; 2707.2 ± 577.0 g). The maternal age, parity, gestational age of the child, and body weight of the child at birth showed no difference between the PPD and AJD patients (*p* > 0.05, Mann–Whitney U test). The interval from childbirth to HRV measurement was 14.9 ± 8.1 days in the PPD patients and 14.8 ± 7.9 days in the AJD patients, showing no group difference (*p* > 0.05, Mann–Whitney U test). For both groups of patients, psychiatric treatments were initiated, including antidepressant medication for PPD. Age-matched women who had not experienced childbirth within the previous year and no history of psychiatric, neurological, cardiovascular, arrhythmic, or metabolic illnesses served as the normal control group (control, *n* = 26, age: 32.3 ± 4.8 years). A Kruskal–Wallis test revealed no statistical differences in mean age among the PDD, AJD, and control groups (PPD vs. control, *p* = 0.38; PPD vs. AJD, *p* = 0.77; AJD vs. control, *p* > 0.99).

### 2.2. Ethical Background

This research was conducted in agreement with the Declaration of Helsinki. All participants in the present study gave us written informed consent. The Institutional Review Board of Shizuoka Saiseikai General Hospital approved the content of the present study (No24-10-03).

### 2.3. Heart Rate Variability Measurement

HRV measurement was conducted at an outpatient clinic of Shizuoka Saiseikai General Hospital. The EPDS-positive women visited the clinic for an examination of their HRV profiles; a wearable device was used to measure electrocardiogram (ECG) signals during a three-behavioral-state paradigm of about a 5 min duration, consisting of the rest (Rest), mental task load (Task), and rest-after-task (After) states, as described below. The data were analyzed at the site of measurement in the outpatient clinic, and the HRV profiles of the women, shown in the Results Section, were obtained immediately after data assessment for several minutes. The present method for HRV measurement during the three-behavioral-state paradigm, thoroughly described below, was preceded by simple and short-term training that enabled the medical staff to conduct the measurement.

After a 5 min adaptation period, each subject was seated on a chair with the wearable ECG device attached to the chest during the measurement (RF-ECG2, GM3, Tokyo, Japan). The device was 40 mm wide, 35 mm long, and 7.2 mm thick, and weighed 12 g. Its attachment to the chest using conventional adhesive electrodes (Blue sensor P-00-S, Ambu A/S, Ballerup, Denmark) caused little distress to the subject. ECG was measured conventionally, with a gain of 10,000 and a time constant of 0.1 s (Bonaly Light, Version 1.2, GMS, Tokyo, Japan). The R-R interval trend was made from the R peaks of the ECG, and the maximum entropy method was employed to assess its variation (MemCalc, Version 1.2, GMS, Tokyo, Japan). The maximum entropy method was capable of analyzing the trend data with a duration of 30 s in the present three-behavioral-state paradigm [34,36]. Fast Fourier Transform (FFT) has been frequently used for power spectrum analyses of trend data, but requires at least 256 data points, which corresponds to about 5 min in a subject with a 60 beats/min heart rate. FFT applied to the present three-behavioral-state paradigm would require approximately 15 min, potentially causing distress in the women. A paradigm with a short duration should be adequate for a point-of-care sensing tool.

The sampling frequency in the present study was 200 Hz, and the R peaks were detected using the software Bonaly Light (Version 1.2, GMS, Tokyo, Japan). For the exclusion of paroxysmal beats, R-R intervals between 273 and 1500 ms were used for analysis. When an interval was omitted, the average of the preceding and following intervals was used. Resampling of the R-R intervals was conducted at the mean heart rate. The number of R-R intervals omitted from the analysis was not available with the present software, but the online monitoring of ECG during the paradigm by the examiner indicated that trains of two or more paroxysmal beats were not observed in all women, and suggested that the omission of R-R intervals did not significantly affect the analysis.

MemCalc [37] was used to calculate low-frequency (LF) and high-frequency (HF) components of the spectrum every 2 s by integrating the power at the corresponding frequency intervals (0.04–0.15 Hz for LF, 0.15–0.4 Hz for HF) [33] for the preceding 30 s period. R-R intervals were also converted to HR (/min). Previous research indicated that HF corresponds to parasympathetic activity in relation to breathing rhythm [32]. Breathing rate was monitored by the examiner, and was found to be between 9 and 24 times a minute (0.15–0.4 Hz), as previously indicated [38]. When the breathing rate was out of this range, the women were asked to modulate their breathing, and to restart the measurement. The present study incorporated frequency-domain HRV indices, including LF and HF, instead of time-domain HRV indices, because each frequency-domain index is related to different components of the autonomic regulation of circulatory control and should be adequate to serve as a parameter to differentiate the PPD, AJD, and control groups using a discriminatory equation, as described below.

We measured ECG in three different states regarding the task load: the rest, task, and rest-after-task states (AMAS, Version 1.0, GM3, Tokyo, Japan). First, the women were asked to be relaxed in a chair for about 60 s (the initial rest state: Rest). Then, they were instructed to perform a random number generation task for 100 s (the task state: Task). Following the task, ECG was measured for a 60 s period in the relaxed condition (the rest-after-task state: After). HRV indices and HR were averaged in the period from 30 s after the beginning of the state to its end so that the data at the beginning of each state that could be affected by the previous state were not included (AMAS, Version 1.0, GM3, Tokyo, Japan, Table 1).

As for the random number generation task, the women vocalized the digits 0 through 9 in an order as random as possible 100 times at a rate of one digit a second. The generation rate was indicated by a click. The women were asked to concentrate on this task, and all of them completed it. The randomness in the generated digit series was evaluated by the random number generation index (RNG index; frequency of same digit pairs), which was calculated according to our previous study [39], and was used to assess task performance.

### 2.4. Statistics

The differences in the HRV and HR indices of each group among the Rest, Task, and After states were examined using a repeated-measures ANOVA with a post hoc Tukey’s multiple comparison test. The differences in the HRV and HR indices in each state among the PPD, AJD, and control groups, as well as the differences in the task performance index (RNG index), were examined using ANOVA with a post hoc Tukey’s multiple comparison test. The differences in the obstetric data between the PDD and AJD patients were examined using a Mann Whiteny U test, and the difference in age among the PPD, AJD, and control groups was checked using a Kruskal–Wallis test (Prism 8, Version 8.4.3, GraphPad Software, San Diego, CA, USA).

### 2.5. Linear Discriminant Analysis

To further establish the usefulness of the HRV and HR indices for differentiating the PPD, AJD, and control women, linear discriminant analysis was employed. The details can be found in our previous publication [35]. For linear discriminant analysis, a linear discriminant equation was made, which was composed of the HRV and HR indices multiplied by the coefficients and a constant (discriminant point) (Equation (1)). For the HRV and HR indices in the equation, their values during the Rest, Task, and After states were used. LF/HF was incorporated because the ratio of LF to HF can give us additional information about HRV. The Task/Rest and After/Rest ratios were employed as they reflect the responsiveness of HRV parameters to task load.

Using the software StatMate V (Version 5.01, ATMS, Chiba, Japan), three equations were determined to discriminate PPD from control, PPD from AJD, and AJD from control. The values of the coefficients in each equation were calculated so that the discriminant score (D-score, Equation (1)) could most effectively discriminate between two groups; D-score is positive when the former diagnosis is supported and negative when the latter one is supported. The sensitivity and specificity in discriminating the former diagnosis from the latter using the D-scores were calculated for the three sets of groups: PPD vs. control, PPD vs. AJD, and AJD vs. control.
(1)$$\begin{array}{c} \text{Discriminant score}\;(\text{D-score})\;=\;\text{a HF}[\text{Rest}]\;+\;\text{b HF}[\text{Task}/\text{Rest}]\;+\text{c HF}[\text{After}/\text{Rest}]\\ +\;\text{d LF}[\text{Rest}]\;+\;\text{e LF}[\text{Task}/\text{Rest}]\;+\;\text{f LF}[\text{After}/\text{Rest}]\\ +\;\text{g LF}/\text{HF}[\text{Rest}]\;+\;\text{h LF}/\text{HF}[\text{Task}/\text{Rest}]\;+\;\text{i LF}/\text{HF}[\text{After}/\text{Rest}]\\ +\;\text{j HR}[\text{Rest}]\;+\;\text{k HR}[\text{Task}/\text{Rest}]\;+\;\text{l HR}[\text{After}/\text{Rest}]\\ -\;\text{discriminant point} \end{array}$$

## 3. Results

### 3.1. HRV and HR Indices

The data for LF, HF, LF/HF, and HR obtained from the control women and the patients with PPD and AJD, along with the ratios of the Task and After scores to the Rest scores (Task/Rest and After/Rest) for each index, are shown in Table 1 (mean ± s.d.).

In Figure 1, the profiles of LF, HF, LF/HF, and HR during the three-behavioral-state paradigm (R: Rest, T: Task, and A: After) are shown. The F and *p* values are shown on the upper part of each profile when the effects of state (Rest, Task, or After) are significant according to the ANOVA results (*p* < 0.05). The upward and downward arrows indicate the statistical significance of increases and decreases, respectively, from the Rest scores revealed by the post hoc Tukey’s multiple comparison test. HF showed a decrease during the Task state (*p* < 0.0001) and returned to the Rest level during the After state (*p* = 0.998) in the control women. In the AJD patients, HF showed a decreasing response during the Task state (*p* = 0.0008), the same as in the control group, but exceeded the Rest level during the After state (*p* = 0.0008). On the other hand, HF in the PPD patients showed no response during the Task and After states. The LF/HF scores and HR in the control women and AJD patients showed the same pattern, with an increment during the Task state (LF/HF, *p* = 0.0001 for the control and *p* = 0.034 for AJD, respectively; HR, *p* < 0.0001 for the control and *p* < 0.0001 for AJD, respectively) and a return to the Rest level during the After state (LF/HF, *p* = 0.065 for the control and *p* = 0.331 for AJD, respectively; HR, *p* = 0.199 for the control and *p* = 0.352 for AJD, respectively). However, LF/HF and HR in the PPD patients showed no significant changes during the three-behavioral-state paradigm.

In Figure 2, the Rest scores as well as the Task/Rest and After/Rest scores among the control women, AJD patients, and PPD patients are presented in the box/whisker plots with Tukey’s format. The F and *p* values at the top indicate that the effects of group (control, PPD, and AJD) were significant according to the ANOVA test (*p* < 0.05). Line connections between the data indicate the presence of statistical differences revealed by the post hoc test. The Task/Rest ratio of HF in the PPD patients was significantly higher than that in the control women (*p* = 0.014) and AJD patients (*p* = 0.006). The decrement in HF during the Task state, which was found in the control women and AJD patients, was attenuated in the PPD patients. In the AJD patients, the After/Rest ratio of HF was significantly higher than that in the control women (*p* = 0.009), indicating an increment in HF after the task was over, which was not observed in the PPD patients. The LF/HF at Rest was significantly higher in the PPD patients than that in the control women (*p* = 0.0002) and AJD patients (*p* = 0.004). The Task/Rest ratio of LF/HF in the PPD patients was lower than the control women (*p* = 0.034). In the AJD patients, the After/Rest ratio of LF/HF was significantly lower than that in the control women (*p* = 0.028), and the PPD patients did not exhibit this profile. The Rest scores, as well as the Task/Rest and After/Rest ratios of LF and HR, were not different among the three groups (control, AJD, and PPD).

### 3.2. Task Performance

The mean RNG index score was 0.337 ± 0.058 in the control women, 0.397 ± 0.126 in the AJD patients, and 0.416 ± 0.104 in the PPD patients (mean ± s.d.). The ANOVA with post hoc Tukey’s multiple comparison did not reveal significant differences in RNG index scores between the control women and AJD patients (*p* = 0.078), between the control women and PPD patients (*p* = 0.077), or between the AJD and PPD patients (*p* = 0.843).

### 3.3. Linear Discriminant Analysis

The three discriminant equations with coefficients (a to l) for the HRV indices (LF, HF, and LF/HF) and HR determined by linear discriminant analysis on each group pair could effectively differentiate the PPD patients from the control women, the PPD patients from the AJD patients, and the AJD patients from the control women. The distribution of D-scores calculated by the linear discriminant equations of the PPD patients vs. the control women, the PPD vs. AJD patients, and the AJD patients vs. the control women is shown in Figure 3, along with the Mahalanobis distance (Md) and *p* values. It is indicated that discrimination using D-scores is possible not only between the PPD patients and control women and between the AJD patients and control women, but also between the PPD and AJD patients (Figure 3). The number of women with positive and negative D-scores, as well as the sensitivity and specificity, are presented in Table 2.

## 4. Discussion

### 4.1. HRV and HR Profiles of PPD and AJD during the Three-Behavioral-State Paradigm

In the present study, HRV and HR measurements were conducted during a three-behavioral-state paradigm composed of the Rest, Task, and After states to assess the responsiveness of autonomic activity to task load and the return to the Rest level. This procedure could reveal the autonomic regulation depending on the behavioral states. In the control women, HF decreased and LF/HF and HR increased during the Task state, and HF, LF/HF, and HR returned to the Rest level during the After state (Figure 1).

This autonomic regulation reflected in HRV and HR was not observed in the PPD patients, showing no significant changes in response to task in addition to a high LF/HF score in the Rest state (Figure 2). In contrast, the AJD patients showed basically the same autonomic regulatory profiles as the control women, although HF during the After state was higher than the Rest score.

As for the responsiveness to task load, the PPD patients showed a higher Task/Rest ratio of HF than the AJD patients and control women and a lower Task/Rest ratio of LF/HF than the control women, indicating that the autonomic regulation during the Task state was different from both the control women and AJD patients. The AJD patients showed no difference from the control women during the Task state. On the other hand, the After/Rest ratios were altered only in the AJD patients, showing a higher After/Rest ratio of HF and lower ratio of LF/HF than those in the control women. The PPD patients did not show abnormal After/Rest ratios (Figure 2).

These observations indicate that the characteristic alteration during the three-behavioral-state paradigm in the PPD patients was the unresponsiveness of HF, reflecting insufficient parasympathetic suppression during the task load, while the baseline parasympathetic activity in the Rest state was not altered. This HRV profile of the PPD patients observed in the present study is partly different from the profile reported in our previous study for patients with major depressive disorder using the same three-behavioral-state paradigm [34]. The reduction in HF during the Rest state found in patients with major depressive disorder in our previous report was not observed in the present study. On the other hand, insufficient suppression during the Task state was common in the PPD patients. It has been reported that there is a large interpersonal difference in HF scores during the Rest state in depression. The present results on PPD suggest that PPD can be considered a subgroup of major depressive disorder with less of a reduction in baseline parasympathetic activity [32]. The profile in the PPD patients may be closer to that in patients with chronic fatigue syndrome [32].

On the other hand, the AJD patients and the control women showed similar profiles of HRV and HR during the three-behavioral-state paradigm, as described above. In AJD, autonomic regulation may be adequately regulated. However, the ADJ patients showed changes in HRV and HR indices when the task was unloaded. During the After state, HF was high and LF/HF was low in the AJD patients, suggesting parasympathetic activation after the task load was over. The use of the three-behavioral-state paradigm could disclose the autonomic disturbances that are characteristic of PPD and AJD. The random generation task was simple and the task load was light. And, the RNG index scores were not different among the three groups, indicating that task performance did not significantly influence the results. It is crucial to assess the functional significance of these HRV derangements in future studies based on the mechanisms of HRV with respect to autonomic activity and blood flow regulation [40,41].

### 4.2. Clinical Applicability of HRV and HR Measurement as a Sensing Tool for Screening PPD

In the present study, discriminant scores obtained from the equation composed of the Rest scores, Task/Rest ratios, and After/Rest ratios of LF, HF, LF/HF, and HR enabled us to differentiate not only the PPD patients from the control women but also the PPD patients from the AJD patients. The AJD patients were also differentiated from the control women. The sensitivity and specificity of discrimination using the discriminant score were high for PPD vs. control, PPD vs. AJD, and AJD vs. control (Table 2). The results indicate the usefulness of this tool in clarifying the diagnosis of mental health issues presented in women during the postpartum period. The differentiation of PPD not only from the control but from AJD is important to adequately start the treatment of PPD. PPD detection after childbirth can be scheduled, with the first step using questionnaires and the second step using HRV measurement. HRV measurement as part of the PPD screening system should improve the effectiveness of questionnaires by increasing their specificity.

HRV measurement is performed using a small wearable ECG device that is placed on the chest and causes little distress to patients. The recording can be performed in a regular outpatient office using a laptop computer. The HRV and HR scores during the three-behavioral-state paradigm can be obtained several minutes after the end of the measurement. When applied clinically, this system can be used as a point-of-care diagnostic aid for PPD and AJD to improve the care of mothers after childbirth.

### 4.3. Limitations and Future Directions

The present study has several limitations. In this study, the EPDS-positive rate among the women after delivery is 4.8%. PPD diagnosed based on the DSM-5 criteria presents a rate of 1.3% in the total sample. The prevalence of PPD is low in comparison with previous reports [5]. This study was conducted in a general hospital situated in a middle-sized city of an Asian country. Cultural differences in the pathology of PPD have been suggested and might have affected the results. Future studies in other environments including socio-demographic factors will be necessary to consolidate the present findings.

The limitations also include the cut-off point of EPDS. The cut-off point of 9 in the present study is low in comparison with other studies [10,42]. The present study used a cut-off point of 9 to widely enroll EDPF-positive women to effectively detect not only PPD patients showing severe symptoms but also those with mild symptoms. A cut-off point of 13 has been frequently used [42]. However, the use of higher cut-off points does not always improve specificity [11]. Future studies using HRV parameters together with different cut-off points of EPDS would be interesting.

Future studies should include EPDS-negative women after childbirth. The present study did not obtain data from these women because of the difficulty in gaining their cooperation to check their autonomic activity when they had no mental health issues. However, the discrimination between PPD and AJD is clinically useful because the treatments for both disorders are different.

As for future directions, HRV measurement during the pregnancy period is important. If the risk factors are made clear by HRV measurement before childbirth, the early detection and prevention of PPD would be possible and should enable adequate intervention for women at risk in perinatal mental health. In the present study, HRV measurement during the three-behavioral-state paradigm effectively discriminated AJD patients and control women who did not show HRV differences during the Rest and Task states. Only the data in the After state were different between the two groups. In comparison with the method obtaining only the Rest state [31], this paradigm is informative with respect to autonomic regulation. Further development of this three-behavioral-state paradigm is important to discriminate PPD from other disorders than AJD, including anxiety disorder and chronic fatigue syndrome, based on previous studies using the same paradigm [34,35].

## 5. Conclusions

Abnormalities in HRV and HR were found not only during the Rest state but also during the Task and After states of the three-behavioral-state paradigm in relation to the diagnoses of PPD and AJD. Discrimination among the control, PPD, and AJD patients was possible using HRV and HR scores. When incorporated into a screening system with questionnaires, HRV and HR measurement will improve the specificity of the screening system and can be used as a point-of-care diagnostic aid for PPD and AJD to improve the care of mothers after childbirth.

## 6. Patents

The contents of the present study are included in the Japanese Patent JP 5,492,247,B2; the European Patent EP 2,862,509,B1; and the U.S. Patent US 8,852,116,B2.

## Figures and Tables

**Figure 1 sensors-24-01459-f001:**
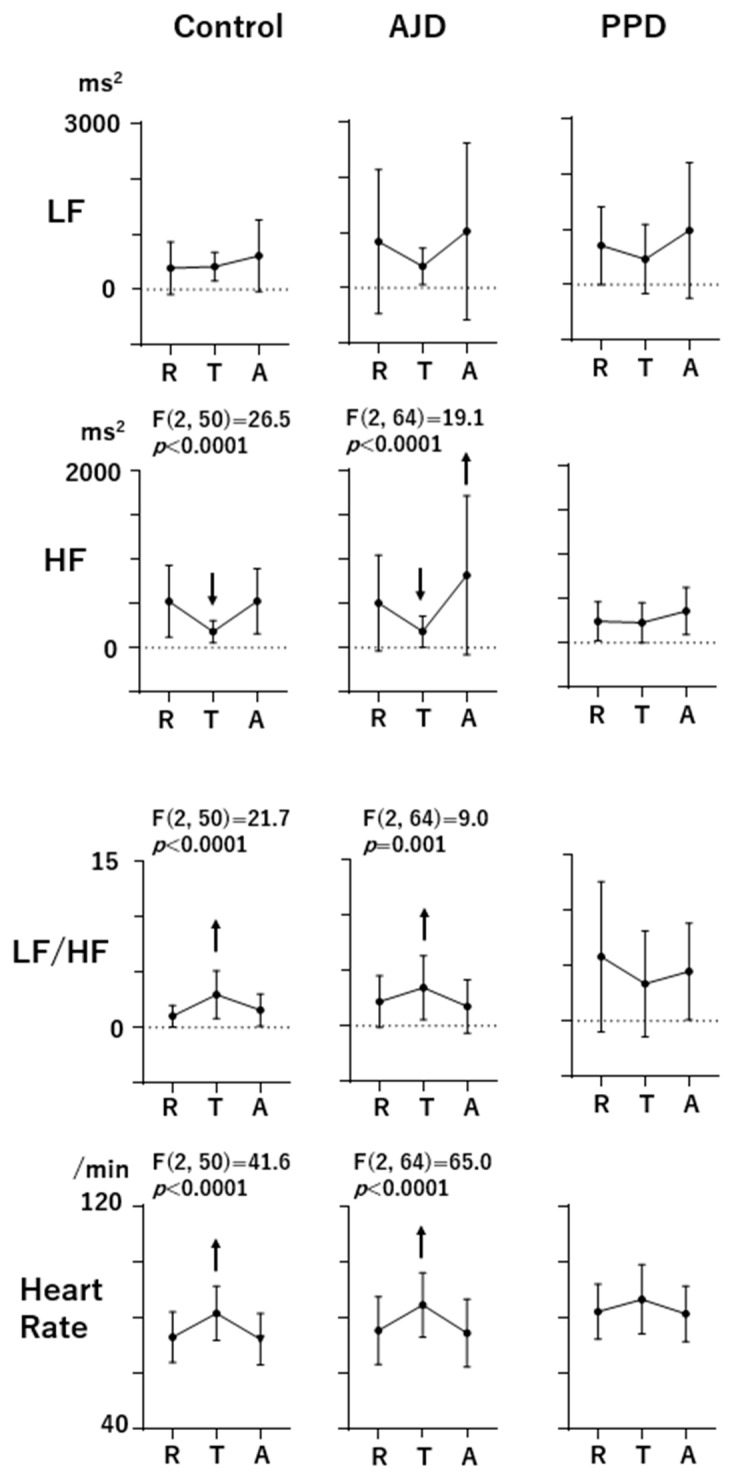
The profiles of LF, HF, LF/HF, and heart rate during the three-behavioral-state paradigm (R: Rest, T: Task, and A: After). The F and *p* values at the top of the profiles indicate that the effects of state (Rest, Task, and After) are significant according to the repeated-measures ANOVA (*p* < 0.05). Upward and downward arrows indicate increment or decrement from the Rest scores, respectively (post hoc Tukey’s multiple comparison test; *p* values are presented in the text).

**Figure 2 sensors-24-01459-f002:**
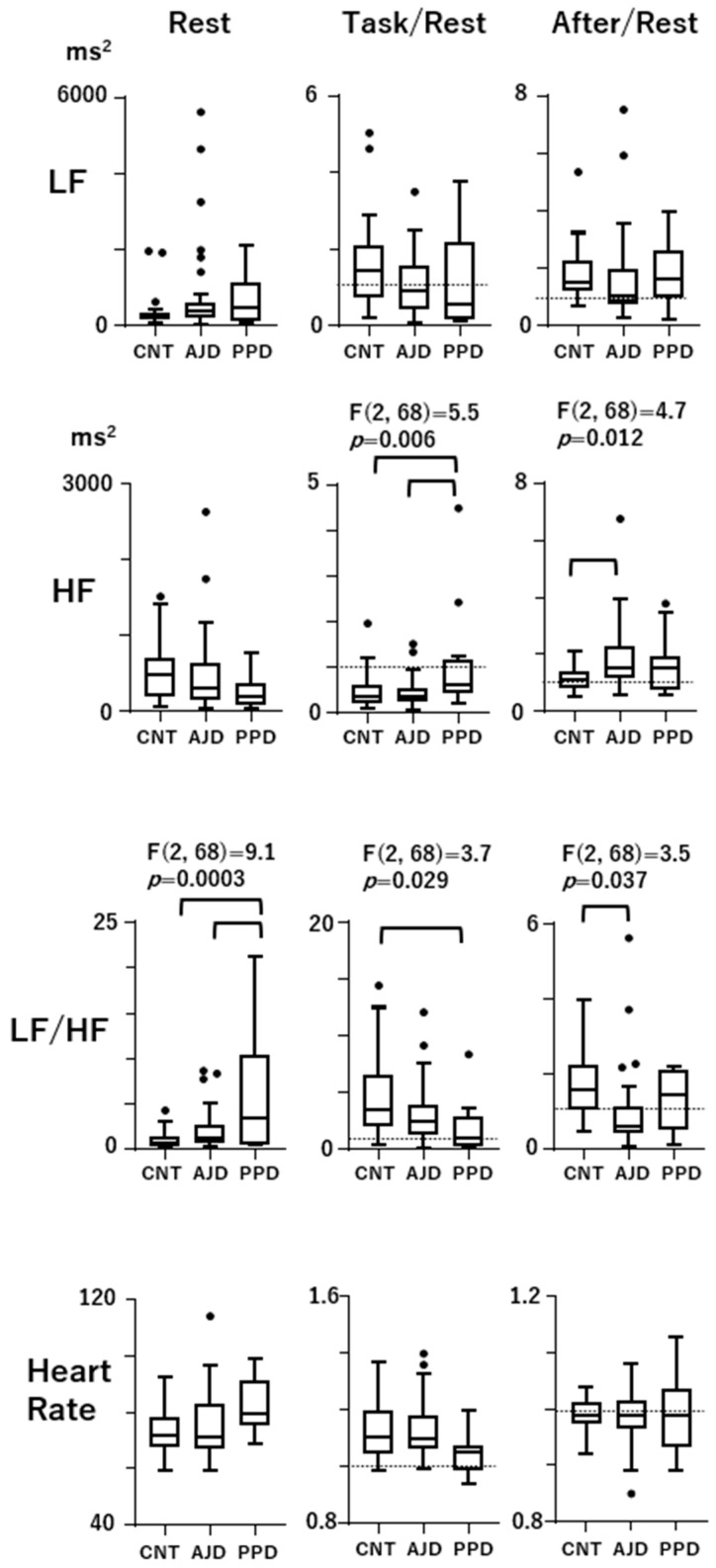
Comparison of the Rest scores as well as the Task/Rest and After/Rest scores among the control, AJD, and PPD patients. The data are presented in box/whisker plots with Tukey’s format. The outliers are shown as black dots. The F and *p* values at the top of the profiles indicate that the effects of group (control, PPD, AJD) are significant according to the ANOVA test (*p* < 0.05). Statistical differences found in the post hoc Tukey’s multiple comparison test are designated by line connections (*p* values are presented in the text). The dashed lines in the Task/Rest and After/Rest figures designate a score of 1, indicating the Rest level.

**Figure 3 sensors-24-01459-f003:**
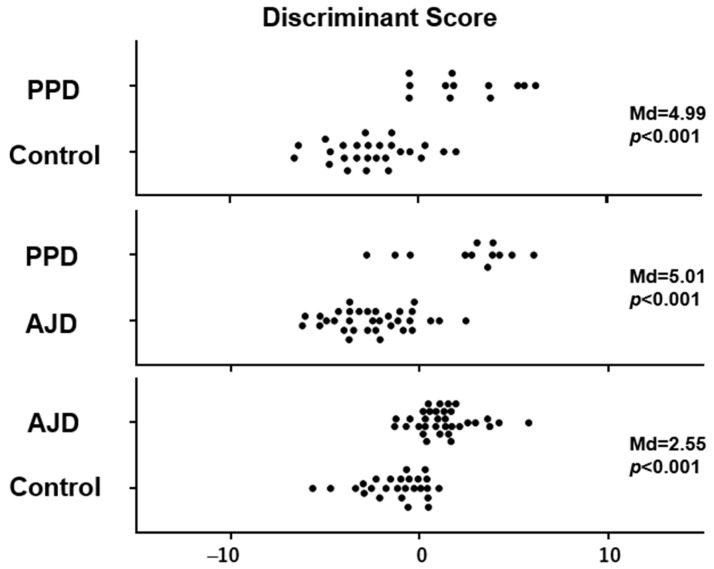
Discriminant scores (D-scores) calculated by the linear discriminant equations for PPD vs. control, PPD vs. AJD, and AJD vs. control. Each filled circle indicates the individual data. Mahalanobis distance (Md) and *p* value are shown on the right side.

**Table 1 sensors-24-01459-t001:** Heart rate variability indices (LF, HF, and LF/HF) and heart rate (HR) during Rest, Task, and After states among the control women (Control), postpartum depression patients (PPD), and adjustment disorder patients (AJD) (mean ± s.d.).

LF	Rest (ms^2^)	Task (ms^2^)	After (ms^2^)	Task/Rest	After/Rest
Control	388.9 ± 474.4	413.3 ± 256.7	612.7 ± 648.0	1.65 ± 1.20	1.82 ± 1.01
AJD	837.4 ± 1300.6	391.5 ± 330.6	1024.8 ± 1599.8	1.02 ± 0.77	1.60 ± 1.52
PPD	707.0 ± 705.3	463.2 ± 629.3	981.1 ± 1224.9	1.12 ± 1.22	1.78 ± 1.11

**HF**					
Control	526.2 ± 405.4	182.0 ± 126.7	528.3 ± 368.0	0.49 ± 0.41	1.15 ± 0.39
AJD	505.0 ± 544.1	183.6 ± 176.1	820.0 ± 897.7	0.46 ± 0.32	1.94 ± 1.25
PPD	243.4 ± 218.6	227.7 ± 226.0	359.2 ± 266.4	1.10 ± 1.22	1.66 ± 1.06

**LF/HF**					
Control	1.04 ± 0.98	2.98 ± 2.16	1.57 ± 1.47	4.63 ± 3.78	1.71 ± 0.90
AJD	2.19 ± 2.34	3.46 ± 2.90	1.76 ± 2.41	3.04 ± 2.74	1.03 ± 1.10
PPD	5.82 ± 6.80	3.37 ± 4.80	4.48 ± 4.37	1.85 ± 2.39	1.34 ± 0.80

**HR**					
Control	73.1 ± 9.1	81.7 ± 9.7	72.4 ± 9.3	1.12 ± 0.10	0.99 ± 0.03
AJD	75.5 ± 12.2	84.7 ± 11.5	74.5 ± 12.3	1.13 ± 0.09	0.99 ± 0.05
PPD	82.3 ± 9.9	86.7 ± 12.5	81.5 ± 10.0	1.05 ± 0.08	0.99 ± 0.07

**Table 2 sensors-24-01459-t002:** Number of women with positive discriminant scores (D > 0) and negative discriminant scores (D < 0).

	D > 0	D < 0	Total
**PPD vs. normal**			
PPD	9	3	12
Normal	4	24	28
Total	13	27	
	*sensitivity*	*specificity*	
	75.0%	84.6%	
**PPD vs. AJD**			
PPD	9	3	12
AJD	3	30	33
Total	12	33	
	*sensitivity*	*specificity*	
	75.0%	90.9%	
**AJD vs. normal**			
AJD	28	5	33
Normal	7	19	26
Total	35	24	
	*sensitivity*	*specificity*	
	84.8%	73.1%	

PPD: postpartum disorder; AJD: adjustment disorder.

## Data Availability

The data that support the findings of this study are available from the corresponding author upon request. The data are not publicly available due to privacy and ethical restrictions.

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
