# Peer review of "Heart Rate Variability Measurement Can Be a Point-of-Care Sensing Tool for Screening Postpartum Depression: Differentiation from Adjustment Disorder"

_sensors, 2024, doi:10.3390/s24051459_

Round 1
Reviewer 1 Report
Comments and Suggestions for Authors
See attached file

Some minor issues are noted in the attached file
Author Response
Dear Reviewer 1
Thank you very much for your valuable comments and suggestions. Following them, we modified the manuscript. The modified parts are highlighted yellow. Below is the list of modifications. We hope that the manuscript is improved.
Sincerely yours,
Toshikazu Shinba
Department of Psychiatry
Shizuoka Saiseikai General Hospital
Comment 1: Use of patients or women instead of subjects is preferable.
Response: Thank you for the suggestion. We replaced ‘PPD and AJD subjects’ by ‘PPD and AJD patients’, and ‘control subjects’ by ‘control women’ throughout the manuscript.
Comment 2: Inclusion of degrees of freedom and actual p values should be added to F values.
Response: Thank you for the important suggestions. All the F values in Figures 1 and 2 are accompanies by degrees of freedom and p values. In addition, p values for post hoc test are presented in the manuscript (lines 250, 255-258, 261-264, 273-285).
Comment 3: Check the format of the references.
Response: Thank you for your comment. We checked and corrected the inadequate format of references.
Additional comments and suggestions:
Response: Thank you for precise and detailed comments and suggestions.
#We modified the manuscript following them throughout the manuscript (lines 20, 23., 27, 43, 87-90, 111, 112, 126, 129, 170-171)
#The description concerning paroxysmal beats was modified as follows; ‘trains of two or more paroxysmal beats were not observed’ (lines 180).
#Modifications following your comments and suggestions (lines 183, 192, 213, 222-230, 377, 390).
#We described that HRV measurement improves the effectiveness of questionnaire ‘by increasing the specificity’ (line 395)
#We added a reference (ref. 42) to discuss the use of higher cut-off point (lines 411-412). #Future studies ‘using HRV parameters together’ with different cut-off points of EPDS are interesting (lines 415-417).
#We deleted a paragraph following your suggestion (line 418)
#We deleted a paragraph following your suggestion (line 420).
Reviewer 2 Report
Comments and Suggestions for Authors
The article is interesting and has an advanced practical approach. But there are controversial things in this article.
Firstly, the obtained result of the HRV response to mental load in the group of women with various disorders does not have significant changes. It is likely that the proposed mental test was not difficult for these subjects.
Second, the Conclusion is formulated as abstracts. It is necessary to reflect the main results of this work in the Conclusion.
Author Response
Dear Reviewer 2
Thank you very much for your valuable comments and suggestions. Following them, we modified the manuscript. The modified parts are highlighted yellow. Below is the list of modifications. We hope that the manuscript is improved.
Sincerely yours,
Toshikazu Shinba
Department of Psychiatry
Shizuoka Saiseikai General Hospital
Comment 1:
Firstly, the obtained result of the HRV response to mental load in the group of women with various disorders does not have significant changes. It is likely that the proposed mental test was not difficult for these subjects.
Response: Thank you for the important comment. In the group of PPD patients, HRV parameters did not respond to task load. We used RNG index to evaluate the performance of the subjects and showed that the performance itself did not significantly affect the results. The following sentences were presented in the Discussion section (line 376) ‘The random generation task is simple and the task load would be light. However, the RNG index scores were not different among the three groups, indicating that the task performance did not significantly influence the results.’
Second, the Conclusion is formulated as abstracts. It is necessary to reflect the main results of this work in the Conclusion.
Response: Thank you for the valuable comment. We deleted some parts of Conclusion and clarified the results (line 436).
Reviewer 3 Report
Comments and Suggestions for Authors
The work addresses an important topic, and I commend your efforts in conducting this research. The study provides evidence that clinical applicability of HRV and HR measurements using Linear discriminant analysis could be valuable tool for screening PPD and enhance the discrimination between PPD and AJD, contributing to the enhancement of postpartum maternal care.
I have one general concern and few minor.
My general concern is regarding the general scientific approach. The authors have a long and fruitful track-record using identical research approach exploiting the HRV and Linear discriminant analysis in similar health conditions. With all due respect to the authors, my opinion is that this kind of approach is lacking the originality and novelty and decreases the value of obtained results for top quality journal as this one. I have read your previous papers and they are not identical but rather very similar. There is no upgrade in applied methods or trying to confirm or compare gathered results with other methods.
Minor comments:
Introduction: Consider providing more context in the introduction to clearly establish the significance of the research problem. Specifically, in the opening part of the paper there is no informations about AJD and why it is important for us discriminate between these two conditions.
Provide additional details on the methodology to enhance the reproducibility of the study.
Please, highlight the key contributions of your study and how they advance the current state of knowledge in the field.
In the limitation part you have addressed some important issues. Can you elaborate little more about Cutt-of point set at 9? Also, it would be interesting to monitor HRV during pregnancy as you stated and do correlation analyses with afterbirth PPD diagnosis.
I believe that addressing these points will significantly strengthen your manuscript. Please consider these comments as constructive feedback aimed at improving the overall quality of your work.
Comments on the Quality of English Language
The language used in the paper is clear, but some sentences are overly complex. Simplifying the language would enhance reader comprehension.
Author Response
Dear Reviewer 3
Thank you very much for your valuable comments and suggestions. Following them, we modified the manuscript. The modified parts are highlighted yellow. Below is the list of modifications. We hope that the manuscript is improved.
Sincerely yours,
Toshikazu Shinba
Department of Psychiatry
Shizuoka Saiseikai General Hospital
The work addresses an important topic, and I commend your efforts in conducting this research. The study provides evidence that clinical applicability of HRV and HR measurements using Linear discriminant analysis could be valuable tool for screening PPD and enhance the discrimination between PPD and AJD, contributing to the enhancement of postpartum maternal care.
I have one general concern and few minor.
Comment 1:
My general concern is regarding the general scientific approach. The authors have a long and fruitful track-record using identical research approach exploiting the HRV and Linear discriminant analysis in similar health conditions. With all due respect to the authors, my opinion is that this kind of approach is lacking the originality and novelty and decreases the value of obtained results for top quality journal as this one. I have read your previous papers and they are not identical but rather very similar. There is no upgrade in applied methods or trying to confirm or compare gathered results with other methods.
Response: Thank you for the important comment. We have been accumulating HRV data using the same three-behavioral-state paradigm to utilize them in the daily medical practice. Based on your comments, we added the following discussion; ‘In the present study, HRV measurement during three-behavioral-state paradigm effectively discriminated AJD patients and control women who did not show HRV differences during the Rest and Task states. Only the data at After state were different between the two groups. In comparison with the method obtaining only the Rest state [31], this paradigm is informative with respect to autonomic regulation. Further development of this three-behavioral-state paradigm is important to discriminate PPD from other disorders than AJD including anxiety disorder and chronic fatigue syndrome based on the previous studies using the same paradigm [34, 35].' (lines 426-434)
Minor comments:
Comment 2:
Introduction: Consider providing more context in the introduction to clearly establish the significance of the research problem. Specifically, in the opening part of the paper there is no information about AJD and why it is important for us discriminate between these two conditions.
Response: Thank you for the important comment and suggestion. In the Introduction section, we added a description of AJD and the importance of discrimination between PPD and AJD to clarify the significance of the present research problem (lines 62-66).
Comment 3:
Provide additional details on the methodology to enhance the reproducibility of the study.
Response: Thank you for the comment. The methodology of HRV measurement is thoroughly presented in the text and added the description as follows; ‘The present method for HRV measurement during the three-behavioral-state paradigm thoroughly described below is simple and short-term training enabled the medical staff to conduct the measurement’ (lines 155-157).
Comment 4:
Please, highlight the key contributions of your study and how they advance the current state of knowledge in the field.
Response: Thank you for the important suggestion. In the Conclusion section, we described the key contributions of this study by stating ‘Discrimination among the control, PPD, and AJD patients was possible using HRV and HR scores. When incorporated into a screening system with questionnaires, HRV and HR measurement will improve the specificity of the screening system and can be used as a point-of-care diagnostic aid for PPD and AJD to improve the care of mothers after childbirth’(line 438).
Comment 5:
In the limitation part you have addressed some important issues. Can you elaborate little more about Cutt-of point set at 9? Also, it would be interesting to monitor HRV during pregnancy as you stated and do correlation analyses with afterbirth PPD diagnosis.
Response: Thank you for the valuable comment. As for the cut-off point, we added discussion concerning the cut-off point of 9 in the present study, as follow ‘The limitation also includes the cut-off point of EPDS. The cut-off point of 9 in the present study is low in comparison with other studies [10, 42]. The present study used the cut-off point of 9 to widely enroll the EDPF-positive women to effectively detect not only the PPD patients showing severe symptoms but those with mild symptoms. The cut-off point of 13 has been frequently used [42]. However, the use of higher cut-off points did not always improve specificity [11]. Future studies using HRV parameters together with different cut-off points of EPDS are interesting’ (lines 411-417). As for the HRV monitoring during pregnancy, we agree with you and would like to start the research in the future.
I believe that addressing these points will significantly strengthen your manuscript. Please consider these comments as constructive feedback aimed at improving the overall quality of your work.
Comments on the Quality of English Language
The language used in the paper is clear, but some sentences are overly complex. Simplifying the language would enhance reader comprehension.
Response: Thank you for the comments. We tried to simplify the description as much as possible.